# LEARNING CLUSTER STRUCTURED SPARSITY BY REWEIGHTING

## ABSTRACT

Recently, the paradigm of unfolding iterative algorithms into finite-length feed-forward neural networks has achieved a great success in the area of sparse recovery. Benefit from available training data, the learned networks have achieved state-of-the-art performance in respect of both speed and accuracy. However, the structure behind sparsity, imposing constraint on the support of sparse signals, is often an essential prior knowledge but seldom considered in the existing networks. In this paper, we aim at bridging this gap. Specifically, exploiting the iterative reweighted $\ell_1$ minimization (IRL1) algorithm, we propose to learn the cluster structured sparsity (CSS) by rewegihting adaptively. In particular, we first unfold the Reweighted Iterative Shrinkage Algorithm (RwISTA) into an end-to-end trainable deep architecture termed as RW-LISTA. Then instead of the element-wise reweighting, the global and local reweighting manner are proposed for the cluster structured sparse learning. Numerical experiments further show the superiority of our algorithm against both classical algorithms and learning-based networks on different tasks.

## 1 INTRODUCTION

Sparsity is an important inherent property that describes the low-dimensionality of signals. Learning the sparse representation of signals or data plays an important role in many applications, such as image restoration (Liu et al., 2018; 2019b), classification (Wright et al., 2009), radar detection (Ahmad & Amin, 2013), medical imaging (Lustig et al., 2007), or black hole image in astronomy (Honma et al., 2016). As a fundamental, sparse recovery or sparse representation (SR) has been substantially investigated in the last two decades due to the emergence of Compressive Sensing (CS) (Zhang & Rao, 2011; Yu et al., 2012; Needell & Tropp, 2009; Yu et al., 2015; Daubechies et al., 2004).

Particularly, the objective of SR is to find a concise representation of a signal using a few atoms from some specified (over-complete) dictionary

$$\boldsymbol{y} = \boldsymbol{A}\boldsymbol{x} + \epsilon \tag{1}$$

with $\boldsymbol{y} \in \mathbb{R}^M$ the observed measurements corrupted by some noises $\epsilon$, $\boldsymbol{x} \in \mathbb{R}^N$ the sparse representation with no more than $S$ nonzero entries ($S$-sparsity) and $\boldsymbol{A} \in \mathbb{R}^{M \times N}$ the dictionary (normally $M \ll N$). Consequently, the ill-posedness of (1) prohibits a direct solution of $x$ by inverting $\boldsymbol{A}$. In the last two decades, such ill-posed inversion problem has been exhaustively investigated in the community of signal processing (Becker et al., 2011; Daubechies et al., 2004; Needell & Tropp, 2009). Many iterative algorithms such as ISTA (Daubechies et al., 2004), FISTA (Beck & Teboulle, 2009; Becker et al., 2011), ADMM (Chartrand & Wohlberg, 2013) and AMP (Donoho et al., 2009) have been proposed to solve it, which can be considered as *unlearnable* approaches since all the parameters are fixed instead of being learned from data.

Recently, inspired by the deep learning techniques, SR problem is instead turn to the data-driven approaches. The seminal work LISTA (Gregor & LeCun, 2010; Sprechmann et al., 2013; Zhang & Ghanem, 2018) and its consecutive variations (Zhou et al., 2018; Moreau & Bruna, 2016; Ito et al., 2019; Sun et al., 2016) have largely improved the SR performance by simply unfolding iterations of existing SR algorithms into limited number of neural network layers, and then training the network through back propagation with huge amount of data. Theoretically (Moreau & Bruna, 2016)

and empirically (Gregor & LeCun, 2010; Sprechmann et al., 2013; Zhang & Ghanem, 2018), the resulted algorithms are with improved accuracy and efficiency comparing to their original counterparts (Giryes et al., 2018).

On the other hand, the structure behind sparsity, imposing constraint on the support of sparse signals, is an important prior knowledge that can be used to enhance the recovery performance (Wang et al., 2014; Prasad et al., 2015; Liu et al., 2018; Yu et al., 2012). Specifically, structures often exhibit as a sharing of zero/nonzero pattern for grouped entries (dependent on each other) of sparse signals. And the cluster structured sparsity (CSS) is one of the most important cases, where the zero/nonzero entries appear in clusters. Since many natural sparse signals tend to have clustered sparse structure, this pattern has been widely exploited in many practical applications such as gene expression analysis (Tibshirani et al., 2005), and inverse synthetic aperture radar imaging (Lv et al., 2014). However, to the best of the authors' knowledge, few of the data-driven SR algorithms have been proposed for structured SR problems. Thus, there is an urgent need for the study of a learnable approach to utilize the structure behind sparse signals.

In this paper, we try to bridge this gap with special attention on cluster structured sparse recovery. We aim at tackling this problem with neural networks which are explainable and fully trainable. As discussed above, LISTA provides an exemplar approach to bridge the connection between SR and NN in an explainable manner. To further consider the clustered structure, we exploit the reweighted iterative soft-thresholding algorithm (RwISTA) (Fosson, 2018) as the prototype of our proposed network. Essentially, the weights in RwISTA act as a latent variable that controls the sparsity pattern of corresponding coefficients. Based on this observation, in addition to unfolding RwISTA as a feedforward network, we recast the reweighting process into a parameterized reweighting block that can inference the inherent dependencies between coefficients and thus promote structures. To this end, a *one-layer fully connected reweighting block* (RW-LISTA-fc) that exploits the global dependencies of all elements is first proposed. And empirically, the adjacency matrix of the FC layer after training with CSS signals reveals local dependence of elements for CSS signals. Consequently, a *two-layer convolutional reweighting block* (RW-LISTA-conv) is then proposed to capture such dependencies, namely the connection of neighboring coefficients. Through exhaustive numerical experiments, the resulted network RW-LISTA is proven on the superiority over existing methods in solving CSS recovery. Our high-level contribution can be concluded as follows:

- We propose a novel supervised approach for cluster structured sparse recovery. Leveraging the iterative reweighting algorithm, we demonstrate that, we can learn the structured sparsity by reweighting adaptively. Furthermore, a deep architecture called RW-LISTA is proposed by unrolling the existing model into deep networks.

- Instead of element-wise manner, we introduce a fully connected (FC) and a convolutional reweighting block in RW-LISTA to promote the structures. Excitedly, the learned adjacency matrix in FC layer reveals local dependency for coefficients of CSS signals, which is suitable to be captured by convolution. Although this work specially focuses on CSS, it gives further insights to other structured problems beyond SR.

- We achieved state-of-the-art performance on several tasks based on both synthesized and real-world data. In these cases, we compare our algorithm with others in different settings. And the result further verifies the effectiveness and superiority of our proposed methods.

## 2 LEARNING CSS BY REWEIGHTING

### 2.1 REWEIGHTING THE $\ell_1$ NORM ITERATIVELY

Sparse recovery is often cast into a LASSO formulation, namely a minimization problem with $\ell_1$ norm constraint used to promote sparsity. To alleviate the penalty on nonzero coefficients, Candes et al. (2008) propose to reweight the $\ell_1$ norm. Formally, consider the following problem:

$$\min_{\boldsymbol{x} \in \mathbb{R}^N} ||\boldsymbol{y} - \boldsymbol{A}\boldsymbol{x}||_2^2 + \lambda \sum_{i=1}^{N} w_i |x_i| \tag{2}$$

where $w_i$ corresponds to the weight associated to $i$th entry of $\boldsymbol{x}$. To reach a more "democratic" penalization among coefficients, Candes et al. (2008) assign the weight $w_i = \frac{1}{|x_i| + \epsilon}$ with $\epsilon > 0$ a

small value. And thus, $\sum_{i=1}^{N} w_i |x_i|$ can be seen as a smooth approximation of $\ell_0$ norm. Through reweighting, different entries are penalized differently. Large weights will discourage nonzero entries in the recovered signal while small weights tend to encourage nonzero entries. Consequently, to improve the estimation, the weights should be determined properly according to the true location of nonzero entries.

In the original paper, Candes et al. (2008) adopts an iterative algorithm (IRL1) that alternates between reconstruction $\hat{x}$ and redefining the weights, which falls into a general form of majorization-minimization algorithm (a generalization of EM algorithm (Dempster et al., 1977)). Later, Fosson (2018) proposed a simple variant algorithm called Reweighted Iterative Soft-Threshold Algorithm (RwISTA), which can be formulated as:

$$\boldsymbol{w}^{(k)} \leftarrow \varphi\left(\left|\boldsymbol{x}^{(k)}\right|\right), \ \ \boldsymbol{x}^{(k+1)} \leftarrow \Gamma\left(\boldsymbol{x}^{(k)} + \frac{1}{L}\boldsymbol{A}^T(\boldsymbol{y} - \boldsymbol{A})\boldsymbol{x}^{(k)}, \frac{\lambda}{L}\boldsymbol{w}^{(k)}\right) \tag{3}$$

where $\Gamma(\cdot, \cdot)$ is an element-wise soft-thresholding function defined as $[\Gamma(\boldsymbol{x}, \theta)]_i = \max(|x_i| - \theta_i, 0) \cdot \text{sign}(x_i)$ for $i = 1, 2, ..., N$. $L$ is often taken as the largest eigenvalue of $\boldsymbol{A}^T\boldsymbol{A}$, $|\cdot|$ applies element-wisely and $\varphi(\cdot)$ is the reweighting function that takes the magnitude of signal as input. Generally, the reweighting function is usually defined element-wisely as:

$$[\varphi(\boldsymbol{x})]_i = g'(x_i), \ i = 1, 2, ..., N \tag{4}$$

where $g(\cdot)$ is a concave, non-decreasing function called merit function. Obviously, if we define $g(x) = x$, then $\varphi(\boldsymbol{x}) = \mathbf{1}$, (3) will be reduced to normal ISTA. However, a popular choice is $g(x) = \log(x + \epsilon)$ with $\epsilon > 0$ a pre-defined parameter. In this case, the weights are determined inversely proportional to the magnitudes of the corresponding coefficients as $w_i = \frac{1}{|x_i| + \epsilon}$. Consequently, if any coefficient $x_i^{(k+1)}$ becomes larger, the corresponding weight becomes smaller, and in the next iteration a weaker penalty will be applied. Since nonzero entries are more likely to be identified with greater magnitude, iterative reweighting algorithm introduces a positive feedback that prevents the overshrinkage of nonzero entries while suppressing the magnitude of zero coefficients.

## 2.2 Learned Iterative Soft-Thesholding Algorithm with Reweighting Block

Although quite successful, there are some reasons that make RwISTA unsuitable for our problem, namely learning to recover cluster structured sparsity (CSS). First and most importantly, the reweighting function is often defined element-wisely, which ignores the structures behind sparsity. In CSS signals, for example, images of MNIST digits, nonzero entries tend to appear in clusters so that the sparsity pattern of the neighboring coefficients are statistically dependent. But such relationship cannot be captured by the element-wise reweighting manner. Besides, RwISTA can be viewed as a recurrent structure with pre-fixed parameters. However, according to recent successful applications of deep learning in sparse representation, learnable parameters that benefit from huge amount of training data will behave significantly better with respect to both convergence speed and recovery accuracy.

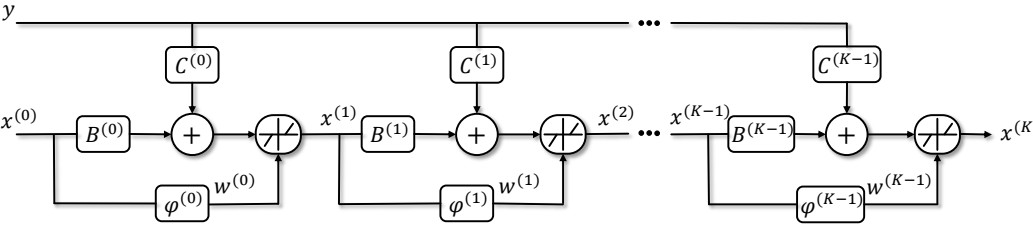

Figure 1: Architecture of our proposed model RW-LISTA.

To address the issues mentioned above, we propose RW-LISTA as shown in Figure 1, which unfolds RwISTA (3) into a finite-length deep neural network. Formally, the forward pass of RW-LISTA can be formulated as:

$$\boldsymbol{w}^{(k)} \leftarrow \varphi^{(k)}\left(\left|\boldsymbol{x}^{(k)}\right|\right), \ \ \boldsymbol{x}^{(k+1)} \leftarrow \Gamma\left(\boldsymbol{B}^{(k)}\boldsymbol{x}^{(k)} + \boldsymbol{C}^{(k)}\boldsymbol{y}, \theta^{(k)}\boldsymbol{w}^{(k)}\right) \tag{5}$$

where $\boldsymbol{B}^{(k)} \in \mathbb{R}^{N \times N}$, $\boldsymbol{C}^{(k)} \in \mathbb{R}^{N \times M}$ and $\varphi^{(k)}(\cdot)$ denotes the paratermized reweighting block in the $k$th layer. The model's architecture is illustrated at Fig. 1. In RwISTA we fix $\boldsymbol{B}^{(k)} = \mathbf{I} - \frac{1}{L}\boldsymbol{A}^T\boldsymbol{A}$, $\boldsymbol{C}^{(k)} = \frac{1}{L}\boldsymbol{A}^T$ and $\theta^{(k)} = \frac{\lambda}{L}$ for all layers. However, now all the parameters in RW-LISTA (including the parameters of the reweighting block $\varphi^{(k)}$) are optimized by minimizing the reconstruction error $||\hat{\boldsymbol{x}} - \boldsymbol{x}^*||_2^2$ through back-propagation to fix the distribution of the data.

Since weights are able to control the sparsity pattern of the coefficients, what we need to do is to determine the weight properly so that they have small values in nonzero locations and significantly greater values elsewhere. It turns out that for structured sparsity, the magnitude of the entry is not the only clue that can be utilized. The dependencies between coefficients should also be carefully considered. Existing works including Yu et al. (2015; 2012) and Fang et al. (2014; 2016) exploit the neighboring dependencies to tackle the problem of CSS recovery, where if the neighbors of one coefficient appears to be nonzero with great magnitude, then the current coefficient is more likely to be nonzero and thus should be reweighted less. It inspires us to build the reweighting block $\varphi$ with respect to the amplitude of coefficients $\boldsymbol{x}$. And the objective is to output weights with small values if the corresponding coefficients belong to some clustered blocks, while with large values if the corresponding coefficients are isolated or small in magnitude. The details of reweighting block construction will be discussed in the next section. And Figure 2 illustrates an example of the output of the proposed reweighting block $\varphi$ (after training).

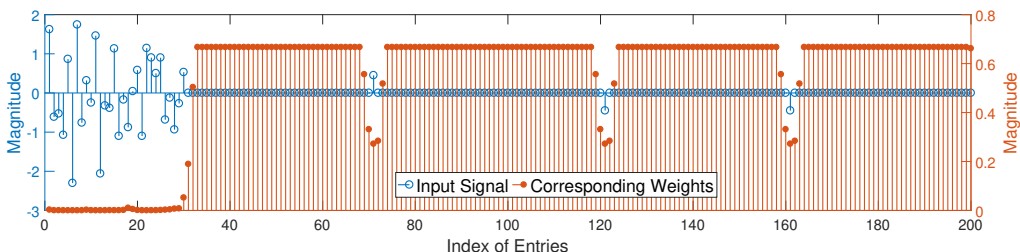

Figure 2: An example of input signal and output of the proposed reweighting block $\varphi$ (after training).

## 2.3 BUILDING THE REWEIGHTING BLOCK

In this section, we discuss the construction of reweighting block. Empirically, we use reversed sigmoid function $\sigma(-x) = \frac{1}{1+e^x}$ as the activation since it's also the derivative of a concave non-decreasing merit function. Traditionally, reweighting is achieved in an element-wise manner: each coefficient will be passed into a derivative of merit function to determine its own weight. This manner introduces a self-dependence between coefficients, and the corresponding reweighting block can be expressed as:

$$\boldsymbol{w}^{(k)} \leftarrow \sigma\left(-t^{(k)}\left|\boldsymbol{x}^{(k)}\right|\right) \tag{6}$$

with $t^{(k)}$ a temperature parameter. Based on previous analysis, the element-wise manner doesn't capture the dependence between elements. Instead, it utilizes the coefficient's self-magnitude as the only clue for reweighting.

To inference the sparsity pattern of CSS signals, we exploit a global dependencies among coefficients of the signal, illustrated at Figure 3 (c). Namely, we build the reweighting block with a full-connection (FC) layer:

$$\boldsymbol{w}^{(k)} \leftarrow \sigma(-\boldsymbol{G}^{(k)}|\boldsymbol{x}^{(k)}|) \tag{7}$$

where the adjacency matrix $\boldsymbol{G} \in \mathbb{R}^{N \times N}$. Different from (6), the FC layer gives the reweighting block a global visual field. Each coefficient is connected to all other elements of the signal. Since each coefficient will pass its influence to other elements through the FC layer and the extent of influence or dependence is measured by the adjacency matrix of FC layer, the learned $\boldsymbol{G}^{(k)}$ in (7) indicates the general sparsity pattern or structure of the signals.

We visualized the matrix $\boldsymbol{G}$ learned on synthesized CSS signals recovery in Fig 4 (c) (refer the details of the experiment in Section 3.1). The learned matrix is notably great in the diagonal areas,

which means each coefficient is highly dependent on its neighbors, but less related to coefficients in a long distance. Generally, it reveals the local dependence of coefficients in CSS signals and is illustrated in Figure 3 (b). Consequently, we further introduce convolutional reweighting block to capture such relationship. In this paper we consider a two-layer convolution:

$$\boldsymbol{w}^{(k)} \leftarrow \sigma\left(-\boldsymbol{v}_2^{(k)} \circledast \text{ReLU}\left(\boldsymbol{v}_1^{(k)} \circledast \left|\boldsymbol{x}^{(k)}\right|\right)\right) \tag{8}$$

where $\boldsymbol{v}_1^{(k)}$ and $\boldsymbol{v}_2^{(k)}$ denote the convolution kernels with the same size and ReLU denotes the rectified linear unit defined as $\max(x, 0)$ (Nair & Hinton, 2010). Zero-padding is added to keep the size unchanged. Since the operation of convolution is local receptive and shift-invariant, its naturally suitable for clustered structure sparsity. Moreover, different kernel sizes will give different visual field for the reweighting block. For example, when the kernel size equals to 1, it will be reduced to element-wise connection manner. Finally, stacked convolutional layers will expand the receptive field and also include more non-linearity in our architecture.

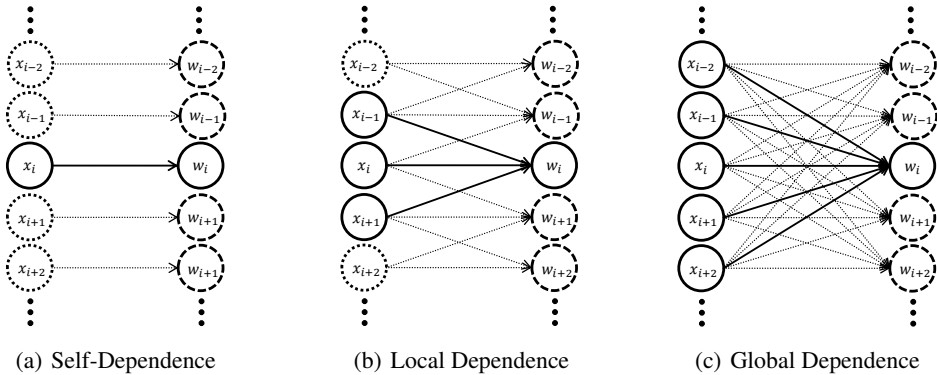

(a) Self-Dependence    (b) Local Dependence    (c) Global Dependence

Figure 3: (a) Self-dependence introduced in the reweighting block, which corresponds to the element-wise manner. (b) Local dependence introduced in the reweighting block, which is achieved by convolution (illustrated by a $1 \times 3$ convolution). (c) Global dependence introduced in the reweighting block, which is conducted by a fully connected layers.

## 3 NUMERICAL EXPERIMENTS

In this section, the results of Monte Carlo simulation experiments on both synthetic and real data are reported for corroborating the above theoretical analysis of the RW-LISTA algorithm. Our experiments contain four parts. The first three parts are based on synthetic data, in which we compare RW-LISTA with both learning-based algorithms and classical CSS solvers to verify the effectiveness of our algorithm. In the final part we consider a real-world application of RW-LISTA: recovery of MNIST digit images.

If not specifically mentioned, the depth of all architecture is set to 12. The sensing matrix $\boldsymbol{A}$ in (1) is randomly generated with each entry independently drawn from a zero-mean Gaussian distribution $\mathcal{N}(0, 1/M)$, and each column is normalized to have unit $\ell_2$ norm. Additive white Gaussian noise (AWGN) is considered and denoted by $\epsilon$, and the signal-to-noise ratio (SNR) in the system is defined as

$$\text{SNR} = \frac{\mathbb{E}[||\mathbf{Ax}||_2^2]}{\mathbb{E}[||\epsilon||_2^2]} \tag{9}$$

The following average normalized mean square error (NMSE) is defined to evaluate the RW-LISTA and compare with existing cluster structured sparse (CSS) recovery algorithms

$$\text{NMSE} = 10 \log_{10} \mathbb{E}\left\{\frac{||\mathbf{x} - \hat{\mathbf{x}}||}{||\mathbf{x}||_F}\right\} \tag{10}$$

where $|| \cdot ||_F$ is the Frobenius norm, and $\hat{\mathbf{x}}$ denotes the recovered CSS vector. Our codes are implemented with Python and MATLAB. Models are built on PyTorch (Paszke et al., 2017) and trained

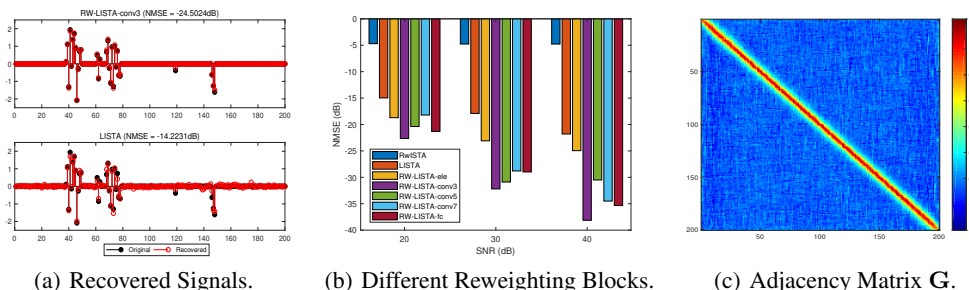

| (a) Recovered Signals. | (b) Different Reweighting Blocks. | (c) Adjacency Matrix $\mathbf{G}$. |

Figure 4: (a) Visualization of recovered signals by LISTA and RW-LSITA. (b) The results of different reweighting blocks on synthetic CSS signals, where RwISTA and LISTA are considered for comparison. (c) Visualization of the matrix of one learned reweighting FC layer.

by Adam Optimizer (Kingma & Ba, 2014). All the experiments are performed on a laptop with Intel Core i7-8750H clocked at 2.20 GHz CPU and a NVIDIA GeForce GTX 1060 GPU.

### 3.1 DIFFERENT REWEIGHTING BLOCKS

In this section, we test different reweighting blocks described in Section 2.3: element-wise, convolution and full-connection. For convolution we consider three kernel sizes, 3, 5 and 7. By setting $N = 200$ and $M = 100$, synthetic CSS signals with non-zero elements are generated following a standard Gaussian distribution.

An example of recovered signal by LISTA and RW-LISTA is illustrated in Figure 4 (a), and the results of recovery NMSEs versus different SNR levels after 12 layers/iterations are shown in Figure 4 (b), where RwISTA and LISTA are used for comparison. It is noted that the proposed RW-LISTA achieves significant performance gain with much lower NMSEs, compared to RwISTA and LISTA. In particular, reweighting with $1 \times 3$ convolution achieves the best performance. Furthermore, the superiority of RW-LISTA can be further enlarged as the SNR level increases. To get a better sense of the pattern learned through the reweighting blocks, the learned weight matrix $\mathbf{G}$ in equation (7) from one layer of the reweighting blocks is visualized, as shown in Figure 4 (c). The weight is noticeably great in the diagonal area, i.e., each coefficient is highly connected to its neighbors. This indicates that the RW-LISTA has successfully learned the clustered structure sparsity. Since such a learned matrix assembles the operation of convolution, in the following we choose to use $1 \times 3$ convolution for the RW-LISTA considering computational cost and recovery performance.

### 3.2 EFFECTIVENESS OF REWEIGHTING BLOCK

The next experiment will focus on verifying the effectiveness of the proposed reweighting block. To achieve this goal, we integrate the reweighting block into LISTA and its successors, i.e., LISTA-CP (Chen et al., 2018), TIED-LISTA as well as ALISTA (Liu et al., 2019a)), and obtain their reweighted versions: RW-ALISTA-CP, RW-TIED-LISTA, and RW-ALISTA. We also include the result of LISTAs with support selection proposed in Chen et al. (2018) for comparison (termed as LISTA-SS, LISTA-CPSS, TIED-LISTA-SS, and ALISTA-SS). As before, we generate synthetic CSS signals with non-zero elements following Gaussian Distribution under $N = 200$ and $M = 100$. The results of NMSE comparison between RW-LISTAs and LISTAs in the case of different noisy conditions are presented in Figure 5 (a)-(d), where all the curves are depicted by running 10000 Monte Carlo trials.

It is clearly observed that the proposed RW-LISTAs with reweighting block achieve significant performance improvement compared to LISTA and its variants, especially under relatively high SNR conditions. This implies that the performance of CSS signal recovery has been certainly improved by adaptively reweighting the coefficients in the thresholding operation of ISTA. It is worthwhile noticing that the role of the reweighting block becomes more obvious when the level of SNR increases.

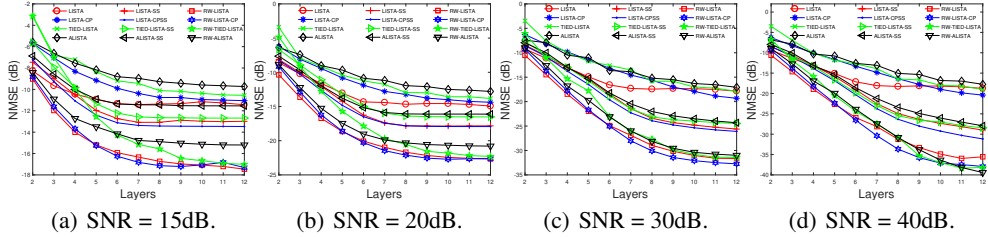

(a) SNR = 15dB.  (b) SNR = 20dB.  (c) SNR = 30dB.  (d) SNR = 40dB.

Figure 5: Comparisons of reweighted LISTAs with LISTAs at different SNR levels.

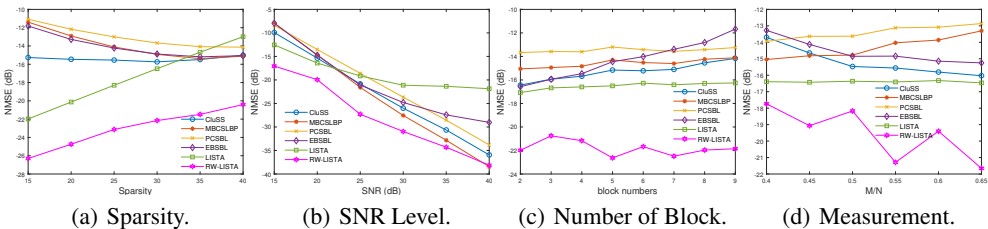

(a) Sparsity.  (b) SNR Level.  (c) Number of Block.  (d) Measurement.

Figure 6: Comparisons of RW-LISTA with state-of-the-art CSS solvers.

### 3.3 COMPARISON OF RW-LISTA TO CLASSICAL CSS SOLVERS

Signal recovery techniques for CSS are various and can be roughly classified into two main categories depending on whether the deep learning is involved. In order to demonstrate the advantage of parameter learning based techniques, synthetic CSS data are generated for evaluating the RW-LISTA in comparison with classical CSS recovery algorithms, including CluSS (Yu et al., 2012), MBCSLBP (Yu et al., 2015), PCSBL (Fang et al., 2014), and EBSBL (Zhang & Rao, 2013). Suppose the sparsity of a $N \times 1$ CSS signal is $S$, i.e., there are $S$ non-zero elements, which consist of $T$ blocks with random sizes and locations. The results are averaged over 500 independents runs.

The effectiveness and robustness of the RW-LISTA are demonstrated in Figure 6 in terms of the following four aspects: i) **Sparsity**: Note that from Figure 6 (a) the RW-LISTA outperforms the others by a large margin, which verifies the theoretical analysis in section 2. The reason lies in that the reweighting block is embedded to improve recovery accuracy while maintaining acceptable computational complexity. ii) **Noise level**: The RW-LISTA is noise-robust and can achieve higher recovery accuracy than those using classical CSS solvers especially in low SNR levels, as shown in Figure 6 (b). Note that the efficient learning via the reweighting block is the key to achieve the desired performance. iii) **Number of blocks**: The number of blocks changes when keeping the sparsity unchanged. It is seen in Figure 6 (c) that the NMSE curve of RW-LISTA slightly fluctuates as the number of blocks increases, and there is a significant performance improvement when compared with existing algorithms. The results in Figure 6 (a) and (c) indicate that the performance degradation mainly results from the change of sparsity rather than the number of blocks. iv) **Number of measurements**: The effectiveness of the RW-LISTA in coping with different numbers of measurements is likewise demonstrated in Figure 6 (d). It is noted that the RW-LISTA gives rise to substantial gain in the NMSE performance for the reason outlined above. Moreover, we find that the curve of the RW-LISTA does not monotonously change with the number of measurements.

As a result, the above features of the RW-LISTA strongly relax the constraints on the sparsity, noise level, blocks and measurements of the CSS signals. The reweighting block is therefore regarded as a unit which is well-matched for LISTA architecture.

### 3.4 EXPERIMENTS ON MNIST DIGITS RECOVERY

To give additional insight of the proposed RW-LISTA into the signals encountered in real-life scenarios, the experiments are carried out on the MNIST dataset, wherein the monochrome images of hand-written numerals and labels exhibit cluster-sparse structure in spatial domain and most of

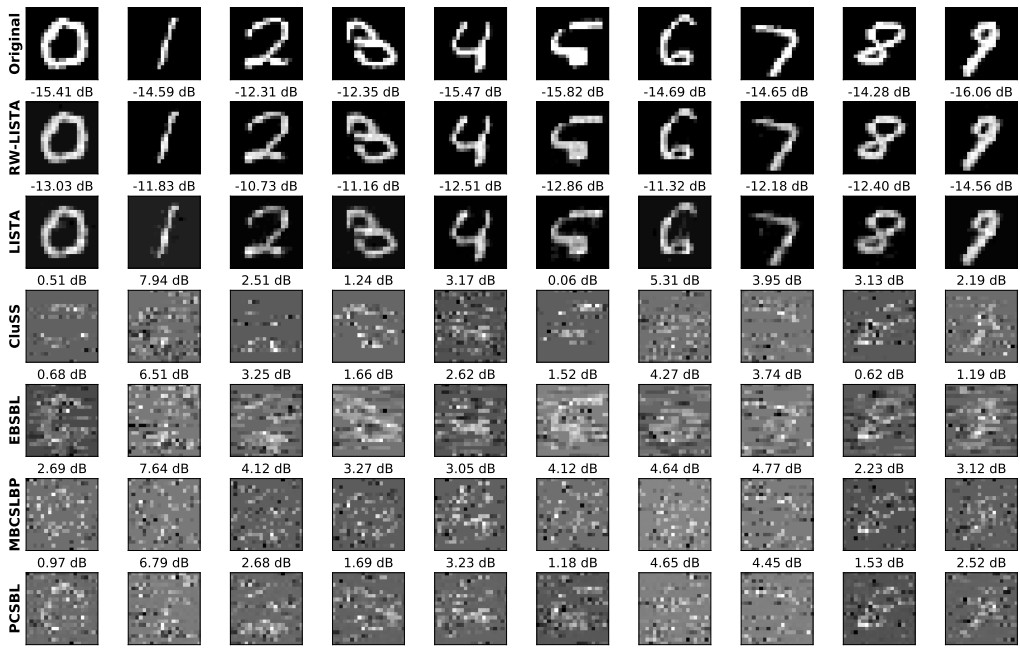

Figure 7: Reconstructed $20 \times 20$ digit images on MNIST dataset using different recovery algorithms with $N = 400$, $M = 200$ and SNR = 5 dB (Rows from up to down: original images, recovered images by RW-LISTA, LISTA, CluSS, EBSBL, MBCSLBP, and PCSBL)

image pixels are zero. Extension of one-dimensional signal recovery to two-dimensional image recovery is straightforward since the two-dimensional image can be transformed into one-dimensional block-sparse signal. In the experiment, We resize the images in MNIST into $20 \times 20$ and normalize the pixel value to $[0, 1]$. Then each image is rasterized into a 400-dimensional vector. We set $N = 400$, $M = 200$ and corrupt the observation with SNR = 5 dB. All the 60000 images in the training set have been used to train RW-LISTA.

An illustrative example of image signal recovery on MNIST dataset is given in Figure 7 with a SNR of 5 dB, where the NMSE of each recovered image is annotated for quantitative comparison. The state-of-the-art CluSS, EBSBL, MBCSLBP, PCSBL, and LISTA algorithms are taken as the reference for performance comparison with the proposed RW-LISTA. It is observed that the results in Figure 6 are in accordance with those in Figure 7, i.e., the RW-LISTA substantially outperforms the others in preserving the cluster property for different image patterns (We choose the numeral images with distinct sparsity and number of blocks). We also note that the reconstructed images by RW-LISTA are very close to the original ones, and this phenomenon can be regarded as a consequence of the use of the reweighting block, which is able to well learn the spatial pattern of hand-written numerals. In contrast, most of state-of-the-art algorithms have the difficulty in recovering the images with real-world spatial structures in such noisy cases largely due to unknown sparse and cluster prior of CSS signals, e.g., cluster locations and the number of clusters.

## 4 CONCLUSION

In this paper, we explore a reweighted deep architecture for the recovery of CSS signals. In the proposed reweighting block, we introduce local dependence of coefficients by convolutional layers, and global dependence by FC layers. Moreover, we choose RwISTA and LISTA as the incarnation of our idea and get a novel architecture termed RW-LISTA. In light of the above analysis of experimental results, the conclusion is reached that the proposed algorithm successfully utilizes the clustered structure of signals and outperforms existing algorithms on different tasks. Besides it also gives further insights into the recovery of other types of structured sparsity based on the reweighting mechanism, which is pending in the future work.

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
