# OpenReview forum: "Learning Cluster Structured Sparsity by Reweighting"
_ICLR.cc/2020/Conference — Reject_

### Official Review · AnonReviewer3 · 2019-10-25
**Official Blind Review #3**

**Rating:** 1

**Review:**

The paper combines deep learning and compressed sensing. Specifically the RW-LISTA algorithm is proposed for cluster-structured sparse recovery, building upon two existing methods: the Reweighted Iterative Shrinkage Algorithm and the LISTA algorithm. The reweighing process is employed to infer the dependencies between coefficients and encourage cluster structure. Strategies for local and global dependence are presented. The approach is evaluated on synthetic and real datasets.

The paper considers and important topic. However the proposed approach is too incremental and the empirical evaluation could also be improved. In addition the presentation needs more work.  Specifically:
- the use of unsupervised vs supervised is misleading. Traditional CS approaches are in fact supervised as they map to a regression problem when both input sensing matrix and response vector are available. The distinction has more to do with the ability (or lack of ability) to learn representations.
- the proposed approach is a straightforward combination of LISTA and RwISTA and the section of global/local dependence regarding cluster-sparse structures is unsurprising.
- Even though signals may exhibit cluster-sparsity, the size of such cluster might vary widely and it is therefore questionable if such patterns can be best captured via connections in the proposed reweighing blocks. Indeed for some blocks wider or lower neighborhoods might be needed to capture various radii of dependence.
- As an alternative to adopting RwISTA it might be pertinent to compare against a counterpart using fused lasso penalty (Tibshirani et al 2005).
- Experiments are limited: a wider variety of block structure with more or less variability in block size etc should be considered. In addition it would be important to compare against vanilla  Rw-ISTA, as one of the classical CSS solvers.
- It is somewhat disappointing that the proposed approach should be better the higher the SNR, where other approaches can do well enough. Ideally we are looking to improve in less favorable conditions of low SNR.


**Experience Assessment:**

I have published in this field for several years.

**Review Assessment: Checking Correctness Of Derivations And Theory:**

I carefully checked the derivations and theory.

**Review Assessment: Checking Correctness Of Experiments:**

I carefully checked the experiments.

**Review Assessment: Thoroughness In Paper Reading:**

I read the paper thoroughly.

---

> ### Author Response · Authors · 2019-11-15
> **Thanks for your valuable and professional opinion**
>
> Q1: "The proposed approach in too incremental." / "The proposed approach is a straightforward combination of LISTA and RwISTA and the section of global/local dependence regarding cluster-sparse structures is unsurprising."
>
> A1: This paper aims at addressing the problem of clustered structure sparse recovery. The main contribution of this paper is the insight of learning (to recover) cluster structured sparsity by reweighting.  Note that traditional reweighting algorithm (candes et al, 2008) was not designed for structured spasity. We expand the reweighting mechanism to structured problems by the power of deep learning. Specifically a reweighting block is proposed to introduce local and global dependencies of the signal's coefficients. And existing algorithms of RwISTA and LISTA are chosen for the incarnation of our idea. However, it should not be limited to RwISTA and LISTA, other algorithms (such as AMP) are also applicable.
>
> Moreover, detailed analysis and experiments is also an important contribution of our work. In section 3 we conduct exhaustive simulations by comparsion to both classical and deep learning based algorithms in different settings. The result of experiments verifies the effectiveness and superioriy of our algorithm especially in noisy cases.
>
> Q2:"Even though signals may exhibit cluster-sparsity, the size of such cluster might vary widely and it is therefore questionable if such patterns can be best captured via connections in the proposed reweighing blocks. Indeed for some blocks wider or lower neighborhoods might be needed to capture various radii of dependence. "
>
> A2: Even though the block size could vary widely, each coefficient is most statistially related to its very nearing neighborhoods. The larger the distances between two coefficients are, the less the dependence is. Considering the variation of block size, a reasonable receptive field of the reweighting block suits best for recovery. This idea is verified by figure 4(b) and (c): in figure 4(b) there is only 5 to 6 elements excitated in each row of the learned adjacency matrix, while in figure 4(c) a reweighting block consists of two 1*3 convolution layers achieves best performance, which is able to couple each coefficient with its neighboring 4 coefficients.
>
> Indeed, some previous works also showed that a relatively reasonable field of connection is enough to handle the variation of block sizes. For example, in PCSBL (Fang et al, 2014) and CluSS (Yu et al, 2015), clustered pattern is captured only via the connection of 2 neighboring coefficients, i.e., $x_i$ is only coupled with $x_{i-1}$ and $x_{i+1}$.
>
> Q3: "The use of unsupervised vs supervised is misleading."
> A3: Thanks for your kind notification. We've revised it and instead use learnable vs. unlearnable to refer the distinction between classical SR solvers and recent deep learning based solvers.
>
> Q4: "A wider variety of block structure with more or less variability in block size etc should be considered. In addition it would be important to compare against vanilla Rw-ISTA, as one of the classical CSS solvers."
> A4: Thanks for you suggestion, we have considered the variety of block structure by changing sparsity and block number of the signal in section 3.3. Please refer to Figure 6(a) and Figure 6(c). Also, we've made comparison with vallina Rw-ISTA Figure (4).
>
> Q5: Ideally we are looking to improve in less favorable conditions of low SNR.
> A5: We've revised the experiments in section 3.3 and set SNR to 5dB, where the difference of LISTA and RW-LISTA is more distincted.
>
> Q6: As an alternative to adopting RwISTA it might be pertinent to compare against a counterpart using fused lasso penalty (Tibshirani et al 2005).
> A6: To our best knowledge, fused lasso is proposed to encourage both sparsity and smoothness of recovered signal. However, in our setting the magnitude of signal's coefficients could vary so we think it's not proper to compare with fused lasso.

---

### Official Review · AnonReviewer1 · 2019-10-26
**Official Blind Review #1**

**Rating:** 6

**Review:**

An approach is proposed to learn sparse representations while preserving some structures in the data.

The idea seems quite nice where we want to learn the structure that induces sparsity instead of simply sparse representations. The idea is to extend a recent algorithm RwISTA by adding reweighting block. The reweighing block changes weights to encode whether coefficients in the model learned by the network are similar or dissimilar to each other. To build the reweighing block convolutional layers are used.

The paper is slightly hard to read due to many typos, and hard-to-read sentences. Also, I think a more intuitive explanation of how the reweighting helps preserve structure is needed. Right now it is very difficult to understand (except maybe for experts working on similar problems?) Regarding the experiments, most of them are run with synthetic cases. It seems like the approach is compared with several recent approaches though showing good results. On the MNIST data, results are shown where the images are recovered from the sparse representation. I did not really see any substantial improvements in performance as compared to say LISTA. Maybe I am misunderstanding what is being evaluated in Figure 7. Also I am not sure how it shows that the sparse representation is learning the underlying structure? Maybe some re-writing is needed to make this clearer.

I think the paper is interesting but needs some polishing to make it easier to read and perhaps some improved experiments in real datasets.


**Experience Assessment:**

I do not know much about this area.

**Review Assessment: Checking Correctness Of Derivations And Theory:**

I did not assess the derivations or theory.

**Review Assessment: Checking Correctness Of Experiments:**

I assessed the sensibility of the experiments.

**Review Assessment: Thoroughness In Paper Reading:**

I read the paper at least twice and used my best judgement in assessing the paper.

---

> ### Author Response · Authors · 2019-11-15
> **Thank you for you careful review and kind notification**
>
> Q1: "The paper is slightly hard to read due to many typos, and hard-to-read sentence."
> A1: Thanks for your careful reading and kind notification of the paper. We'll fix these typos and meticulously proofread  the paper.
>
> Q2: "I am not sure how it shows that the sparse representation is learning the underlying structure?"  / "Also, I think a more intuitive explanation of how the reweighting helps preserve structure is needed."
> A2:  On the one hand, the experiments show that RW-LISTA significantly outcomes LISTA and classical CSS solvers in clustered structuer sparse recovery, revealing that the proposed algorithm has successfully (learned to) utilize the structure priori for better recovery. On the other hand, as shown in Figure 2, the learned reweighting block favors cluster pattern of the signal while depressing isolated coefficients.
>
> Q3: "I did not really see any substantial improvements in performance as compared to say LISTA" /
> "Perhaps some improved experiments in real datasets."
> A3: Thanks for the constructional idea. In different noist conditions, the performance of LISTA against RW-LISTA on MNIST dataset is (measured by NMSE):
> | SNR            |       5 dB      |  10dB   	      |   20dB       |
> | LISTA         |    -12.51dB  |   -15.75dB   |   -23.53dB |
> | RW-LISTA  |   -14.05dB   | -17.17dB     |  -24.23dB  |
> It shows that the superority of RW-LISTA against LISTA is more clear under lower SNR conditions, so we've changed the SNR in section 3.3 to 5 dB.

---

### Official Review · AnonReviewer2 · 2019-10-28
**Official Blind Review #2**

**Rating:** 3

**Review:**

Paper extends LISTA by introducing learned re-weighting for the problem of sparse signal recovery. Paper combines the insights from RW-ISTA (a re-weighted iterative algorithm with fixed parameters and LISTA, a learned iterative algorithm without re-weighting.

Results show that:
(a) On synthetic data, performance of Rw-LISTA is superior to many variants of LISTA for various SNRs
(b) On synthetic data, Rw-LIST is better than classical approaches when SNR, Sparsity and other factors are varied. Why is LISTA not compared under all these different variations?
(c) On MNIST, Rw-LISTA is much better than non-learned approaches, but seems very close to LISTA.

It strikes me that authors make all evaluations based on NMSE. However, motivation for CSS is that there is structure in the recovered signal — however no comparison of the recovered structure is made. While it is true, that is the signal is perfectly recovered, it would follow the structure from this data was obtained, however no such guarantees can be made for non-zero errors.

I would also like to see what happens whens a pure learning based approach (such as denoting auto-encoders) is used to recover the signal. Do they perform worse than Rw-LISTA?

At present, I think the paper doesnot meet the standard of ICLR submissions. However, if authors address my concerns, I am happy to change the rating.

Minor Comments:
“considered as unsupervised approaches since all the paramters are fixed instead of learning from data.” — this is incorrect. A significant human intuition went into design of these systems. While one can argue that even NNet architecture design requires human intuition, the trend is towards using less domain-specific architectures.

**Experience Assessment:**

I have read many papers in this area.

**Review Assessment: Checking Correctness Of Derivations And Theory:**

N/A

**Review Assessment: Checking Correctness Of Experiments:**

I carefully checked the experiments.

**Review Assessment: Thoroughness In Paper Reading:**

N/A

---

> ### Author Response · Authors · 2019-11-15
> **Thanks for you constructional review**
>
> Q1: "Motivation for CSS is that there is structure in the recovered signal — however no comparison of the recovered structure is made. While it is true, that is the signal is perfectly recovered, it would follow the structure from this data was obtained, however no such guarantees can be made for non-zero errors. "
>
> A1: This paper is discussing cluster structured sparsity (CSS) recovery, and the main goal is to utilize the structure priori knowledge to help the recovery of the signal. So we focus more about the signal itself and only meaured our algorithm by NMSE. However,  to make this paper more convicing, we've tested support error rate (SER) of LISTA and RW-LISTA. Specifially, let T(x) denote the support of x and n denote its length, we define SER as $\frac{[T(x)\cup T(x^*) - T(x)\cap T(x^*)]}{n}$. On synthesized data, the comparison of LISTA and RW-LISTA is:
> | SNR            |  15dB    |  20dB     |  25dB    |   30dB   |  35dB   | 40dB    |
> | LISTA         |  0.2327  |  0.2234  |  0.2323  |  0.2466 | 0.2504  | 0.2436  |
> | RW-LISTA  |  0.0282  |  0.0175  |  0.0119  |  0.0076 | 0.0059 | 0.0059  |
>
> Q2: "Why is LISTA not compared under all varialtions in section 3.2?"
> A2: Thanks for your kind notification. Privously we doesn't consider LISTA in section 3.2 because it has been compared in section 3.1. Now we have include LISTA in section 3.3. Please refer to Figure 6 of the paper.
>
> Q3: "I would also like to see what happens whens a pure learning based approach (such as denoting auto-encoders) is used to recover the signal. Do they perform worse than Rw-LISTA? "
> A3: Thanks for you valuable opinion. There've been some work using pure learning based approaches for compresive sensing. For example, Mousavi et al. (2015) has applied a three layer stacked denoising autoencoder (SDA) for compressive sensing. However, in the original paper Sigmoid is used as the activation of each layer, which is not suitable for recovery of signal with gaussian distribution. Based on our simulation, SDA performs quiet bad on synthesized data with its nonzero coefficients following stardard gaussian distribution , but behaves a little better on MNIST dataset. The result on MNIST dataset is:
> | SNR            |       5 dB      |    10dB         |   20dB       |
> | SDA            |    - 8.08dB   |  -8.56dB      |  - 8.88dB   |
> | LISTA         |    -12.51dB  |  -15.75dB    |   -23.53dB |
> | RW-LISTA  |   -14.05dB   | -17.17dB     |  -24.23dB  |
> Still, the performance of SDA is not as good as deep architectures unfolded by iterative algorithm (LISTA).
>
> Q4:"'considered as unsupervised approaches since all the paramters are fixed instead of learning from data.' — this is incorrect".
> A4: Thanks for your kind remind. We found the use of supervised vs. unsupervised misleading. So we have corrected it and instead use learnable vs. unlearnable to denote the distinction between classical SR solvers and recent deep learning based solvers.

---

### Decision · Program_Chairs · 2019-12-19

**Decision:**

Reject

**Comment:**

The paper is proposed a rejection based on majority reviews.